# Genetic Control of Effective Seedling Leaf Rust Resistance in *Aegilops biuncialis* Vis. Accessions from the VIR Collection

**DOI:** 10.3390/plants13162199

**Published:** 2024-08-08

**Authors:** Maria A. Kolesova, Lev G. Tyryshkin

**Affiliations:** Federal Research Center N.I. Vavilov All-Russian Institute of Plant Genetic Resources (VIR), Bolshaya Morskaya, Str. 42-44, 190000 Saint Petersburg, Russia; markolesova@yandex.ru

**Keywords:** *Aegilops biuncialis*, leaf rust, seedling resistance, inheritance, phytopathological test, molecular markers

## Abstract

Leaf rust (caused by *Puccinia triticina* Erikss., *Pt*) is a severe foliar disease of cultivated wheat worldwide. Severe development of the disease results in significant losses in seed yield and quality. Growing immune varieties is the most rational method for *Pt* control in terms of effectiveness and ecological safety. However, the gene pool of cultivated wheat is very narrow for seedling *Pt* effective resistance genes, which hampers breeding for this trait. One of the well-known methods to broaden genetic diversity for resistance is the introgression of highly effective genes from wild relatives into the genomes of cultivated wheat. The *Aegilops* L. species have been proven to be perfectly suited for this purpose. No gene for *Pt* resistance has been transferred to wheat from *Aegilops biuncialis* Vis. (Lorent’s goatgrass) up to now. Previously, we selected eight accessions of the species from the VIR (N.I. Vavilov All-Russian Institute of Plant Genetic Resources) genebank that showed a perfect level of resistance to leaf rust. In this research, we studied the genetic control of resistance using hybridological, phytopathological, and molecular analyses. According to the F_1_–F_3_ hybrid evaluation results, each accession possesses one dominant gene for *Pt* resistance, and genes in different accessions are allelic or very tightly linked. Phytopathological test clone analysis showed that this gene is not identical to *Lr9*, *Lr19*, *Lr24*, *Lr39*, and *Lr47*, which are effective against *Pt* populations in some areas of Russia. This conclusion was partially supported by the results of the identification of DNA markers specific to these genes in bread wheat. Thus, we identified one dominant gene (temporarily symbolized as *LrBi1*) for effective seedling *Pt* resistance; it is recommended for introgression to cultivated wheat via interspecific hybridization.

## 1. Introduction

Brown (leaf) rust (the pathogen is *Puccinia triticina* Erikss. (*Pt*)) ranks among the most prevalent and destructive diseases of cultivated wheat, including *Triticum aestivum* L. [1,2,3]. Yield losses from *Pt* development are usually lower than those from other rusts (stem and stripe ones), but *Pt* causes more significant annual damage due to its higher frequency and broader distribution [4]. Yield losses from *P. triticina* can reach up to 50% in severe cases [5] or more [6].

Cultivation of rust-resistant crop varieties is regarded as the most efficient and eco-friendly strategy for rust management [1,3,4,6,7,8,9,10]. The success of such varieties’ creation is obviously determined by the presence of genes for effective resistance [8,9], preferably those not previously used in breeding processes [10]. As with other diseases, the widespread deployment of varieties with the same genes for leaf rust resistance (*Lr*) inevitably leads to the emergence and multiplication of pathogen races (genotypes) capable of overcoming resistance, resulting in severe affection of previously resistant varieties.

The *T*. *aestivum* genetic reservoir is extremely narrow for effective *Pt* resistance. For example, all samples with a high level of resistance to leaf rust at the seedling stage from the N.I. Vavilov All-Russian Institute of Plant Genetic Resources (VIR, Saint Petersburg, Russia) are protected only by five oligogenes, *Lr9*, *Lr19*, *Lr24*, *Lr39*, and *Lr47* [10,11,12], which have already been used widely in wheat breeding. As a result, they have lost their effectiveness in many agricultural regions [13,14,15,16,17,18,19,20,21,22,23,24,25,26], including those of the Russian Federation [27,28,29,30,31,32,33]. Therefore, expanding the *T*. *aestivum* gene pool with effective resistance to *P. triticina* is a critical objective [11].

There are several methods to broaden wheat genetic diversity for resistance to diseases, such as the search for new genes among landraces, the induction of mutations, and genetic engineering. However, one of the best methods to achieve this urgent task is the introgression of highly effective genes from wild relatives into the genomes of cultivated wheat [4,6,34], with the *Aegilops* L. species being perfectly suited for this goal [35,36,37].

The annual Lorent’s goatgrass (*Aegilops lorentii* Hochst., Mediterranean *Aegilops*) *Ae. biuncialis* Vis. (tetraploid, genome composition UUM^b^M^b^, origin—spontaneous hybrid of *Ae. umbellulata* Zhuk. × *Ae. comosa* Sm. in Sibth. et Sm.) is a rich source of agronomically valuable genes [38], including effective resistance to detrimental foliar diseases. Accessions of this species resistant to powdery mildew, stripe, and leaf rusts have been previously described [38,39,40]. Despite this, no *Lr*-genes have been introgressed into *T*. *aestivum* from *Ae. biuncialis* up to this moment. The only example of *Ae. biuncialis*-derived resistance in synthetic wheat is the description of AD 6 (*T. turgidum* L. + *Ae. biuncialis*) as resistant under field conditions in Ukraine [41]. However, a substantial number of *Pt* resistance genes derived from others U-genome *Aegilops* species have been described: *Lr9* and *Lr76* from *Ae. umbellulata* and *Lr54*, *Lr57*, *Lr58*, *Lr59*, and *Lr62* from *Ae. kotschyi* Boiss., *Ae. geniculata* Roth, *Ae. triuncialis* L., *Ae. peregrina* (Hack. in J. Fraser) Maire et Weiller, and *Ae. neglecta* Req. ex Bertol., respectively [42,43].

Among them, gene *Lr9* is still highly effective in many wheat-growing regions [11,26], but virulence to the gene has been frequently recorded in the United States [23], the Czech Republic [17], Egypt [13], Iran [25], and in some regions within the Russian Federation [28,30,33]. Gene *Lr57* is not effective in different regions of Russia [12,32]. Gene *Lr62* has been reported as effective against a wide range of western Canadian and South African *Pt* pathotypes [44], but it is of no interest in wheat breeding due to the deleterious effects of the translocations with it [45]. Races from North America virulent to *Lr58* at different stages of plant growth have been reported [46].

Eight *Ae. biuncialis* accessions from the VIR genebank were previously evaluated as possessing a perfect level of seedling *Pt* resistance [10,47]; notably, they were not affected by leaf rust at the adult (flag-leaf) stage at artificial inoculation [47] and during five years under natural epiphytotic conditions in the Northwest regions of the Russian Federation (not published). These accessions theoretically could be of great value in wheat breeding for this trait. Understanding the genetic control of resistance is crucial for preventing the same gene introgression from different goatgrass accessions during the labor-intensive process of crossing with *T. aestivum* and subsequent selections. Prior knowledge of the genetics of resistance in *Ae. biuncialis* will facilitate gene/genes’ introgression into cultivated wheat species. Therefore, the general purpose of this work is to reveal the genetic control of effective seedling *Pt* resistance in eight *Ae. biuncialis* entries from the VIR collection.

## 2. Results

### 2.1. Parental Resistance

Under laboratory conditions seedlings of eight accessions of *Ae. biuncialis* (k-1145, k-2892, k-2900, k-2531, k-3006, k-2452, k-4157, and k-4195) confirmed [10,47] their perfect resistance to complex populations of the leaf rust pathogen representing a mixture of samplings from three distant Russian regions (infection type IT 0) (Table 1). The k-3003 accession exhibited a high level of susceptibility to infection (IT 3) (Figure 1).

### 2.2. Genetic Control of Seedling Resistance to Pt

#### 2.2.1. Hybridological Analysis

##### The Number of Genes for *Pt* Resistance

Overall, we produced 36 biparental crosses to investigate the inheritance of *Pt* resistance in *Ae. biuncialis* to complex *Pt* populations. The seedlings of the F_1_ hybrids of eight crosses—*Ae. biuncialis* resistant entries × susceptible k-3003—were highly resistant (IT 0) to the complex population of *P. triticina*, indicating the dominant character of trait inheritance.

In segregating populations, only highly resistant (IT 0) and highly susceptible plants (IT 3) were observed; there were no intermediate infection types. In F_2_, the observed segregations for *Pt* resistance in all crosses did not contradict the expected ratio for trait inheritance by a dominant allele of a single gene (ratio of resistant plants to susceptible plants—3:1). However, for all combinations, the experimental data also indicated the possible control of resistance by one dominant and additionally two complementary recessive genes (49:15) or by a recessive and two complementary dominant oligogenes (43:21). For the combinations of k-1145, k-2452, k-2531, k-3006, k-4157, and k-4195 crossed with susceptible k-3003, the segregations did not contradict the expected ratios under the control of resistance by one dominant and two complementary (dominant and recessive) genes (51:13). For the combinations of k-2531 and k-4195 crossed with the susceptible k-3003, the segregation did not contradict the hypothesis that they have two genes—one recessive and one dominant (13:3) (Table 2).

Segregations for resistance in F_3_ generations corresponded only to those theoretically expected under monogenic control of the resistance to *P. triticina*. In all combinations, the observed phenotypic ratios deviated significantly (*p* < 0.007) from the expected ones under the hypothesis that any of the eight accessions would have two or three genes for *Pt* resistance (Table 3).

##### Allelic Relationship of *Pt* Resistance Genes in Goatgrass Accessions

To determine the identity of the resistance genes, all eight resistant *Ae. biuncialis* accessions were crossed in an incomplete diallelic scheme; in the F_2_ generation, segregations for resistance were absent (Table 4). The χ^2^ values in the case of the presence of two dominant independently inherited genes for resistance (segregation 15 R:1 S) were in limits of 6.13–7.2 for all F_2_ hybrids (*p* = 0.0083–0.0148). These data definitely indicate the identity or very close linkage of the resistance oligogenes in the goatgrass accessions under study.

#### 2.2.2. Phytopathological Analysis

Leaf segments from eight *Ae. biuncialis* entries, which showed resistance to *Pt* complex populations, were infected with 25 pathogen monopustule isolates virulent individually to one of the bread wheat accessions with individual genes *Lr9*, *Lr19*, *Lr24*, *Lr39*, and *Lr47* [48]. All goatgrass accessions showed resistance to all isolates (IT 0); thus, these entries evidently cannot be defended solely by these single genes. We cannot exclude the presence of the functional alleles of these genes in the accessions under study, but in this case, the accessions should have additional genes for effective resistance, which is in evident contradiction with the hybridological analysis results (each accession has one gene for the trait).

#### 2.2.3. PCR Analysis

The DNA markers linked to the genes for resistance *Lr9* [49], *Lr19* (from *Thinopirum ponticum* (Podp.) Barkworth and D. R. Dewey) [50], *Lr24* (from *Th. ponticum*) [51], *Lr39* (*Ae. tauschii* Coss.) [24], and *Lr47* (from *Ae. speltoides* Tausch) [52] in bread wheat were used to identify these genes in eight *Ae. biuncialis* accessions. These genes were highly effective against the complex leaf rust populations in the present study (see Pathogen Material 4.2). The *Ae. biuncialis* accessions under study had no amplicons after PCR with primers Gb (*Lr19*), SCS1302_607_ (*Lr24*), GDM35 (*Lr39*), and PS10 (*Lr47*); amplicons were visualized only in the corresponding bread wheat control genotypes. After PCR with specific primers to the molecular marker SCS5_550_, specific for *Lr9*, the amplified DNA fragments of size 550 bp were detected not only in the positive wheat control (‘Thatcher’ NIL *Lr9*) but in entries k-2900 (Table 1, Figure 2) and k-3003, too.

## 3. Discussion

Leaf rust can cause significant yield losses in cultivated wheat [4,5,6]. Even if the yield is not affected, disease development leads to a decrease in grain quality [4,53]. Although various methods (agronomical and chemical) have been established to reduce *Pt* development, it is well understood that cultivating resistant varieties is the most effective and environmentally friendly method for rust control [6,7,8,9,10]. To breed such varieties, it is necessary to have sources of new genes for resistance [10,11].

The diversity of bread wheat for effective seedling *Pt* resistance genes is very low; after an extensive evaluation of over 5000 accessions from the VIR genebank, a significant number of resistant entries were identified. However, all of them, based on the results of hybridological analysis and phytopatological tests, had only five genes—*Lr9*, *Lr19*, *Lr24*, *Lr39*, and *Lr47* [10,11]. Although these genes were previously highly effective, virulence to genes *Lr9* [13,17,18,21,23,25,28,30,33], *Lr19* [13,14,17,19,27,29,33], *Lr24* [15,16,20,22,23,27,29], *Lr41* [24,26,27,32], and *Lr47* [31,32] has been detected in many regions of the world. This generally means that breeders do not possess highly effective oligogenes for creating varieties of cultivated wheat capable of resisting modern leaf rust populations.

Therefore, expanding the genetic diversity of bread wheat for effective *Pt* resistance is a crucial goal to provide breeders with new genes for the trait [10]. One of the methods to achieve this actual task is the introgression of highly effective genes from wild relatives into the *T. aestivum* genome [34]. The *Aegilops* species related to cultivated wheat are perfectly suited to solve this problem [10,34,35,36,37,38]. The genus *Aegilops* possesses a large number of genes for resistance to different pathogens [54], including those for *Pt* resistance. For example, out of the five genes earlier identified as being highly effective in Russia [10,11,12,48], three were transferred from *Aegilops*: *Lr9* from *Ae. umbellulata*, *Lr39* from *Ae. tauschii*, and *Lr47* from *Ae. speltoides* [42].

Many entries of different *Aegilops* species from the VIR collection were screened for seedling *Pt* resistance, and a relatively large number of resistant entries were found [10,11,55]. It is important to note that not all identified resistant accessions necessarily possess novel genes for resistance. For example, all highly resistant *Ae. tauschii* and *Ae. cylindrica* host entries from the VIR genebank are protected by only the gene *Lr39*, *Ae. umbellulata* by *Lr9* [10,56]. Knowledge of the trait genetic control in newly identified resistant accessions will allow us to avoid transferring the same genes into the genome of *T. aestivum*, as observed for *Lr39* (=*Lr41*). This gene was introgressed into the *T. aestivum* genome from at least five *Ae. tauschii* accessions of various geographical origins and one *Ae. cylindrica* accession. Hybridological and molecular analyses have shown that *Lr39* and *Lr41* genes are identical or allelic [24].

The goatgrass *Ae. biuncialis* was found to be a rich source of agronomically valuable traits [38], including tolerance to abiotic [57,58,59,60] and biotic stresses [38,39,40,61]. We earlier identified *Ae. biuncialis* accessions with a high level of seedling resistance to *Pt* from the Northwest region of the Russian Federation [10,47]. In this study, their resistance has been confirmed to complex populations of the pathogen, representing a mixture of samples from three distant regions of Russia. Notably, there are no *Lr*-genes introgressed into cultivated wheat from this species. Knowledge of the genetic control of resistance allows us to avoid the laborious and time-consuming transfer of the same gene from different entries of *Ae. biuncialis* to the cultivated wheat species. Traditionally, the study of the genetic control of disease resistance has been based on hybridological analyses (revealing the numbers of genes for the trait and their identity) and phytopatological tests (the allelic relationship of newly identified genes with previously described ones) [62]. Nowadays, DNA markers are also often used for the direct determination of the presence of known genes [62].

As a result of hybridological analysis (crosses of resistant accessions with susceptible ones), we found that each *Ae. biuncialis* accession under study had one gene for resistance, with the dominant allele determining the trait. The absence of segregation in hybrids obtained from crosses of resistant accessions with each other proved the identity of the genes in all accessions under investigation (or very tight linkage).

Allelic relationships of newly identified genes for *Pt* resistance and previously known ones in bread wheat could be easily studied by analyzing hybrids from crosses of new sources with sources of “old” genes (e.g., the ‘Thatcher’ near-isogenic line (NIL) in the case of *Pt* resistance). However, this approach is not applicable in the case of distant species. Therefore, to study possible allelism of the identified *Ae. biuncialis* genes and known genes for effective *Pt* resistance, with a phytopathological test being used. All the *P. triticina* genotypes, virulent to one out of the five genes for effective *Pt* resistance, were not virulent to seedlings of the eight *Ae. biuncialis* accessions. Therefore, their resistance cannot be determined by single genes *Lr9*, *Lr19*, *Lr24*, *Lr39*, and *Lr47* (their presence or the presence of orthologous genes can theoretically be supposed due to the existence of a distant common ancestor of the species-donors of the genes). The results of the test-clone analyses could not precisely indicate the absence of these genes if the accessions have additional effective genetic factors. This hypothesis, however, is in direct contradiction with the hybridological analysis results (one gene for *Pt* resistance in each accession).

Molecular markers are widely used for identifying *Lr* genes not only in *T. aestivum* [63,64,65,66] but also in its relatives [41,67,68]. In this work, we did not observe amplification products after PCR of DNA from *Ae. biuncialis* accessions with four pairs of primers (Gb, SCS1302_607_, GDM35, and PS10) specific to *T. aestivum* for effective genes *Lr19*, *Lr24*, *Lr39*, and *Lr47*, respectively. An amplification product of the *Lr9* gene marker SCS5_550_ was found in one resistant and one susceptible *Ae. biuncialis* accession. However, based on the results of the hybridological analysis and phytopathological test, this gene is likely absent in the sample k-2900. Moreover, this gene cannot be present in the susceptible k-3003 accession. Thus, the result of DNA marker SCS5_550_ identification is in direct contradiction with resistance evaluation and analysis of segregation in hybrid populations. Similarly, the absence of a relationship between the presence of amplification products and functional alleles of genes *Lr39* and *Lr9* in *Aegilops* was previously shown for *Ae. tauschii* and *Ae. umbellulata*, respectively [56].

No previous work has been performed on the genetics of *Ae. biuncialis*‘s effective resistance to leaf rust, with one exception [69]. Two accessions of the species, namely k-2044 and k-2900, were found to be resistant to the disease under field conditions at the adult stage of ontogenesis, and the latter accession was evaluated as highly resistant at the seedling stage to six *Pt* pathotypes. Both accessions had the gene *Lr9* based on the results of the analysis of the presence of a DNA marker for this gene, and k-2044 additionally had *Lr19* (from *Th. ponticum*). One of these accessions was present in our work, and the hybridological analysis and phytopathological test evidently showed the absence of *Lr9*. It should be emphasized that the combination *Lr9 + Lr19* (both effective against leaf rust inoculum used for inoculation) was additionally postulated in three accessions of *Ae. crassa* Boiss., one of *Ae. juvenalis* (Thell.) Eig, and one of *Ae. peregrina* [69], with the first four accessions being highly susceptible to the disease. From our viewpoint, this indicates the impossibility of using DNA markers for reliable *Lr* gene postulations in wild relatives of cultivated wheat [56], as well as in bread wheat [70].

Besides resistance to *Pt*, it was previously shown that the *Ae. biuncialis* entries k-1145, k-2900, k-3006, k-2452, and k-2531 are characterized by a high level of resistance to powdery mildew [71]. Therefore, these accessions are sources of group resistance to two harmful diseases.

The identified *Ae. biuncialis* accessions are of interest for bread wheat improvement. We propose using the temporary symbol *LrBi1* for the gene for *Pt* resistance of these accessions.

## 4. Materials and Methods

### 4.1. Goatgrass Accessions

Genetic determination of leaf rust resistance was studied in eight accessions of *Ae. biuncialis*, namely, k-1145, k-2452, k-2531, k-2892, k-2900, k-3006, k-4157, and k-4195. All of these accessions are from the VIR genebank; their origin is presented in Table 5 [72]. These accessions were previously identified as exhibiting an extremely high level of *Pt* resistance at the seedling and flag-leaf stages of plant growth [10,47]. The accession k-3003 was used as a *Pt* susceptible parent in crosses and control of successful infection. The field appearance of two goatgrass accessions and their seedling reaction to *Pt* infection are shown in Figure 1.

### 4.2. Pathogen Material

The phytopathogen *P. triticina* was sampled on the foliage of susceptible bread wheat genotypes in three different regions of Russia (Leningrad region, Ulyanovsk district, and South Dagestan) several times during wheat vegetation. Different *Pt* samples were combined to create complex populations. Under laboratory conditions, *P. triticina* was maintained on leaf segments of the universally susceptible cv. ‘Leningradka’ (spring bread wheat, Soviet Union, VIR catalog number k-47882) [11].

The complex *Pt* populations under our experimental conditions were virulent to seedlings of *T. aestivum* lines with forty five *Lr* genes (*Lr1*, *Lr2a*, *Lr2c*, *Lr3bg*, *Lr10, Lr11*, *Lr12*, *Lr13*, *Lr14a*, *Lr14b*, *Lr15*, *Lr16*, *Lr17*, *Lr18*, *Lr20*, *Lr21*, *Lr22a*, *Lr22b*, *Lr23*, *Lr25*, *Lr26*, *Lr28*, *Lr29, Lr27 + 31, Lr32*, *Lr33*, *Lr34*, *Lr35, Lr36*, *Lr37*, *Lr38*, *Lr43, Lr44*, *Lr45*, *Lr46*, *Lr48*, *Lr49*, *Lr51*, *Lr52*, *Lr57*, *Lr60*, *Lr63*, *Lr64*, and *Lr67*); in the Leningrad region, only five genes (*Lr9*, *Lr19*, *Lr24*, *Lr39*, and *Lr47*) remained highly effective against leaf rust [12,48]. Lines with these genes were resistant to the complex populations used for goatgrass accessions’ seedling inoculations.

Twenty-five monopustule *Pt* isolates virulent to the effective genes were previously selected [73]; five monopustule isolates were virulent to each gene. These isolates were maintained on the cv. ‘Leningradka’ leaf segments.

### 4.3. Hybridological Analysis

#### 4.3.1. Crossing of the Accessions

All accessions were vernalized before being sown in the soil. The seeds were sterilized in peroxide (3%) for 5 min and then washed in sterile water for 10 min. They were germinated in Petri dishes on cotton wool for three days, and the dishes were then placed in a fridge (+4 °C) for 45–60 days. After this period, the seedlings were gently sown in the field or in greenhouse pots.

Crossings of *Pt*-resistant accessions with susceptible k-3003 and with each other were carried out at the Pushkin Experimental Field (the Russian Federation, Leningrad Region) and in a greenhouse in 2020–2021.

Anthers from the ears of the resistant plants were manually removed, and the ears were covered with isolators made from transparent parchment. Pollination with pollens from k-3003 was carried out using the twell method [74] 2–4 days after emasculation, depending on the weather conditions and the maturity of the pistils’ stigmas.

#### 4.3.2. Evaluation of Resistance

Seeds of parents and hybrids were sown in plastic cuvettes on cotton wool pads and then transferred to a light chamber (3000 lux, temperature–20 °C, constant light).

After 12–15 days, the cuvettes were placed in plastic boxes, and the seedlings (1–2 leaf stage) of the hybrid and parental plants were sprayed with *Pt* spore suspensions (uredospore concentration-3 × 10^4^ mL^−1^). The boxes were covered with polyethylene film and then with a plastic lid and placed in darkness. The next day, the film and lid were removed, and the boxes were returned to the light chamber. One box contained cuvettes with plants of two hybrid populations and three parental goatgrass accessions.

Infection types were scored on the 12–15th day after inoculation. The infection types (IT) 0, 0;, 1, 2, single pustules, and X were related to resistance (R) and susceptibility (S) [75].

After the first evaluation, susceptible plants were removed, and the remaining seedlings were infected a second time to confirm their resistance. Infection types were scored as described above. For segregation analysis, data from two successive inoculations were combined.

#### 4.3.3. Statistical Analysis

The chi-squared (χ^2^) test [76] was utilized to compare the observed segregations for resistance with theoretical Mendelian segregations. To automate the process, an original program for χ^2^ calculation was created in MS Excel 365 version 2407 [62].

### 4.4. Phytopathological Analysis

Leaf segments (0.8–1.0 cm) from eight *Ae. biuncialis* accessions were infected with single *Pt* monopustule isolates under laboratory conditions.

The test clones were virulent individually to seedlings of one of the ‘Thatcher’ near-isogenic lines or varieties of bread wheat with five effective resistance genes, ‘Thatcher’ NILs *Lr9*, *Lr19*, *Lr24*, KS90WGRC10 (*Lr39*), and cv. ‘Pavon’ *Lr47* (see Pathogen Material 4.2). Five monopustule isolates virulent to each gene were selected earlier [12,48,56,73].

Seven days after inoculation of the samples under study, infection types were scored according to the above-mentioned scale. If a certain *Ae. biuncialis* accession showed a susceptible reaction (IT 3) to inoculation with at least one *Pt* isolate virulent to a specific gene for resistance, it could be regarded as likely possessing this gene for resistance [11,56,64,65,73].

### 4.5. Analysis of Amplicons

Goatgrass DNA was extracted from the leaf tissues of 10-day seedlings according to the protocol developed by Edwards et al. [77] with modifications by Dorokhov and Klocke [78]. In our study, DNA markers closely linked in bread wheat to leaf rust resistance genes *Lr9* [49], *Lr19* [50], *Lr24* [51], *Lr39* [24], and *Lr47* [52] were analyzed (Table 6). All primers for the markers were purchased in Evrogen (Moscow, Russia).

The PCR was performed in thermocycler MyCycler Thermal Cycler (BioRad, Hercules, CA, USA). According to the original protocols developed by the scientists who created these markers:-*Lr9*—2 min at 92 °C; 35 cycles (1 min at 94 °C, 1 min at 4 °C, 1 min at 72 °C); final extension for 7 min at 72 °C [49];-*Lr19*—5 min at 94 °C; 40 cycles (30 s at 94 °C, 30 s at 60 °C, 1 min at 72 °C); final extension for 5 min at 72 °C [50];-*Lr24*—2 min at 95 °C; 35 cycles (1 min at 94 °C, 1 min at 60 °C, 1 min at 72 °C); final extension for 7 min at 72 °C [51];-*Lr39*—4 min at 94 °C; 30 cycles (30 s at 94 °C, 30 s at 55 °C, 30 s at 72 °C); final extension for 5 min at 72 °C [24];-*Lr47*—3 min at 94 °C; 7 cycles of touchdown (30 s at 94 °C for; in every cycle the annealing temperature steps down (70, 69, 68, 67, 66, 65, and 64 °C) for 30 s, 30 s at 72 °C); 35 cycles (30 s at 94 °C, 30 s at 63 °C, 30 s at 72 °C); final extension for 7 min at 72 °C [52].

PCR products were electrophoresed in 2.0% agarose (Suzhou Yacoo Science, Suzhou, China), stained with ethidium bromide (0.5 mg L^−1^) solution, and visualized using GelDoc Go Imaging System (BioRad, Hercules, CA, USA). The GeneRuler™ 100 bp Plus DNA Ladder (Fermentas, Waltham, MA, USA) was used to estimate the size of PCR amplified fragments. The ‘Thatcher’ NILs *Lr9*, *Lr19*, *Lr24*, KS90WGRC10 (*Lr39*), and cv. ‘Pavon’ *Lr47* were used as positive controls.

## 5. Conclusions

Leaf rust (*Pt*) is a widespread disease of cultivated wheat, and cultivating resistant varieties is one of the best methods for *Pt* control. To create such varieties, a set of effective genes for resistance should be available to crop breeders. Unfortunately, the genetic diversity of bread wheat is very narrow for effective *Pt* resistance. Moreover, even existing genes can lose their effectiveness due to genetic changes in the pathogen’s virulence (genetic erosion) and/or due to phenotypic changes in virulence under variable abiotic conditions (phenotypic erosion) [73]. Consequently, replenishment of the breeding process with new effective *Pt* resistance genes is a constant and important goal of pre-breeding research. Interspecific hybridization is considered an important approach to achieve this goal, with species of *Aegilops* being of particular interest. Due to technical difficulties, low crossability, and the costs of gene transfer between distant species, the genetic control of resistance should be studied to avoid the introgression of the same gene from different *Aegilops* entries. In this study, the genetic control of effective *Pt* seedling resistance in eight *Ae. biuncialis* accessions from the VIR genebank was investigated with hybridological analysis, phytopathological testing, and DNA-based gene identification. All accessions were shown to possess the same dominant gene for resistance, which is not identical to the genes *Lr9, Lr19*, *Lr24*, *Lr39*, and *Lr47*, which are effective in some regions of the world, including several regions of Russia. We suggest the temporary symbol *LrBi1* for this gene. This gene is of practical interest for cultivated wheat breeding for effective *Pt* resistance. Moreover, five *Ae. biuncialis* accessions were also found to be highly resistant to powdery mildew and could be considered sources of group resistance to two foliar diseases.

## Figures and Tables

**Figure 1 plants-13-02199-f001:**
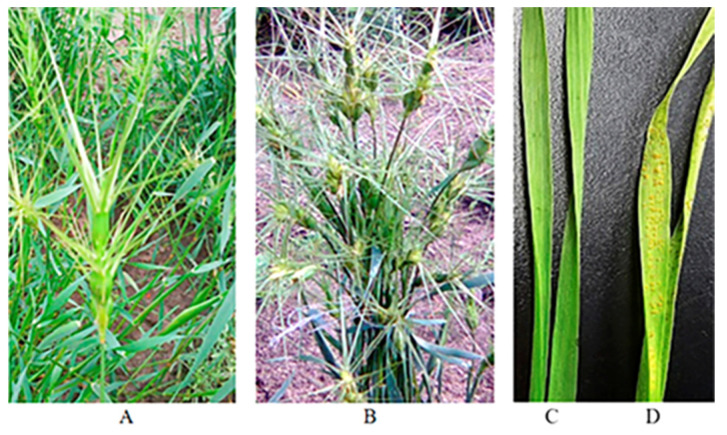
Appearance of *Aegilops biuncialis* accessions in the field: (**A**) resistant k-1145; (**B**) susceptible k-3003; (**C**) rust development on k-1145; and (**D**) on k-3003 accessions.

**Figure 2 plants-13-02199-f002:**
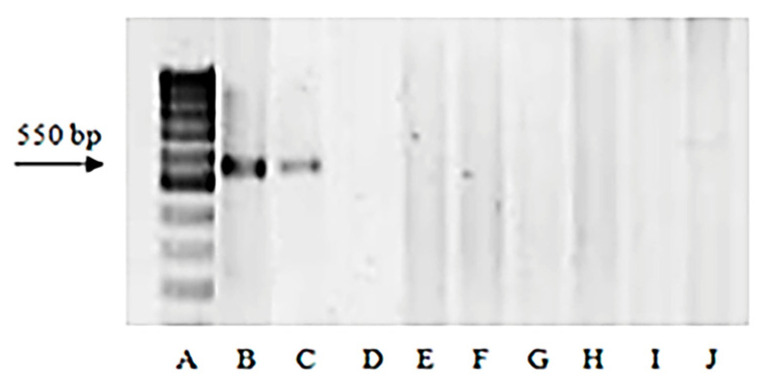
Amplificons of DNA marker SCS5_550_. (**A**) 100 bp DNA ladder; (**B**) bread wheat NIL *Lr9*; *Aegilops biuncialis* accessions resistant to *Pt* (**C**) k-2900; (**D**) k-1145; (**E**) k-2892; (**F**) k-2531; (**G**) k-2452; (**H**) k-3006; (**I**) k-4157; and (**J**) k-4195.

**Table 1 plants-13-02199-t001:** Characteristics of *Aegilops biuncialis* accessions for seedling *Pt* resistance and the 550 bp amplicon synthesis after PCR with the SCS5_550_ primer.

VIR Catalogue №, k-	Infection Type after Inoculation with		Amplified Fragment Presence
	Complex Populations	Clones Virulent to *Lr9*	
1145	0	0	−
2892	0	0	−
2900	0	0	+
2452	0	0	−
2531	0	0	−
3006	0	0	−
4157	0	0	−
4195	0	0	−
3003	3	3	+
‘Thatcher’ NIL *Lr9*	0	3	+

**Table 2 plants-13-02199-t002:** Segregations for *Pt* resistance in the F_2_ hybrids of *Aegilops biuncialis* accessions.

Hybrid Combination	Ratio of Phenotypes R:S *		χ^2^	*p*
	Observed	Expected		
k-1145 × k-3003	73:28	3:1 (1 dominant gene)	0.40	0.53
		49:15 (1 dominant gene + 2 complementary recessive genes)	1.03	0.31
		43:21 (1 recessive gene + 2 complementary dominant genes)	1.19	0.28
		51:13 (1 dominant + 2 complementary: recessive and dominant)	3.43	0.06
k-2892 × k-3003	79:33	3:1	1.19	0.31
		49:15	2.27	0.13
		43:21	0.57	0.45
k-2900 × k-3003	78:32	3:1	0.98	0.32
		49:15	1.96	0.16
		43:21	0.69	0.41
k-2452 × k-3003	75:27	3:1	0.12	0.73
		49:15	0.52	0.47
		43:21	1.86	0.17
		51:13	2.39	0.12
k-2531 × k-3003	70: 25	3:1	0.09	0.76
		49:15	0.44	0.51
		43:21	1.82	0.18
		51:13	2.12	0.15
		13:3 (1 dominant gene + 1 recessive gene)	3.57	0.06
k-3006 × k-3003	69:26	3:1	0.28	0.60
		49:15	0.82	0.37
		43:21	1.28	0.26
		51:13	2.92	0.09
k-4157 × k-3003	72:28	3:1	0.48	0.49
		49:15	1.16	0.28
		43:21	1.05	0.31
		51:13	3.65	0.06
k-4195 × k-3003	72:25	3:1	0.03	0.86
		49:15	0.29	0.59
		43:21	2.18	0.14
		51:13	1.79	0.18
		13:3	3.14	0.08

* R—resistant; S—susceptible.

**Table 3 plants-13-02199-t003:** Segregations of F_3_ *Aegilops biuncialis* families for *Pt* resistance.

Hybrid Combination	Ratio of Families R:RS:S *		χ^2^	*p*
	Observed	Expected		
k-1145 × k-3003	14:29:15	1:2:1 (1 gene)	0.03	0.99
		19:38:7 (1 gene + 2 complementary genes)	13.27	0.001
k 2892 × k-3003	20:43:18	1:2:1	0.41	0.81
		19:38:7	10.65	0.004
k-2900 × k-3003	21:48:24	1:2:1	0.29	0.87
		19: 38:7	21.32	0
k-2452 × k-3003	13:30:16	1:2:1	0.32	0.85
		19:38:7	16.01	0
k-2531 × k-3003	21: 45:26	1:2:1	0.59	0.75
		19:38:7	28.40	0
		7:8:1 (2 genes)	80.54	0
k-3006 × k-3003	16:35:18	1:2:1	0.13	0.94
		19:38:7	16.33	0
k-4157 × k-3003	27:72:24	1:2:1	3.73	0.16
		19:38:7	10.76	0.005
k-4195 × k-3003	26:50:20	1:2:1	0.92	0.63
		19:38:7	9.67	0.007
		7:8:1	38.85	0

* R—resistant; RS—segregating for the resistance; S—susceptible.

**Table 4 plants-13-02199-t004:** Segregations for *Pt* resistance (R:S *) in the F_2_ from *Aegilops biuncialis*-resistant accession crosses.

Parents	♀						
♂	k-2892	k-2900	k-2452	k-2531	k-3006	k-4157	k-4195
k-1145	92:0	105:0	95:0	103:0	96:0	97:0	102:0
k-2892	-	99:0	98:0	98:0	107:0	95:0	108:0
k-2900	-	-	107:0	100:0	105:0	99:0	107:0
k-2452	-	-	-	97:0	108:0	94:0	106:0
k-2531	-	-	-	-	102:0	102:0	100:0
k-3006	-	-	-	-	-	100:0	102:0
k-4157	-	-	-	-	-	-	107:0

* R—resistant; S—susceptible.

**Table 5 plants-13-02199-t005:** Sample data of accessions of *Aegilops biuncialis* from the VIR collection under study.

Catalog No. kk-	Origin	Collecting Site	Collecting Year	Longitude	Latitude
1145	USSR	Nagorno-Karabakh, Gadrut district, Banazur village, surroundings	1971	E 47	N 39
2452	Greece	Attica, Athens, surroundings	1986	E 23	N 37
2531	USSR	Crimea, Ai-Donil village, surroundings, Yalta district	1983	E 34	N 44
2892	Bulgaria	Varna region, Albena-Batovo road, hills	1990	E 27	N 43
2900	Bulgaria	Burgas region, Yablochevo village, Dolgopol-Aitus road	1990	E 27	N 42
3003	USSR	Crimea, Bulganak Hill Field	1988	E 36	N 45
3006	USSR	Crimea, Arabat Bay coast, quarries	1988	E 35	N 45
4157	Greece	Thessaly	2007	E 21	N 39
4195	Greece	Unknown	2007	-	-

**Table 6 plants-13-02199-t006:** Characteristics of DNA markers used to identify *Pt* resistance genes.

Gene	Chromosome Localization	PCR Marker	Primer Sequence (5′-3′)	Fragment Size (bp)
*Lr9*	6B	SCS5_550_	TGC GCC CTT CAA AGG AAG TGC GCC CTT CTG AAC TGT AT	550
*Lr19*	7D	Gb	CAT CCT TGG GGA CCT C CCA GCT CGC ATA CAT CCA	130
*Lr24*	3D	SCS1302_607_	CGC AGG TTC CAA ATA CTT TTC CGC AGG TTC TAC CTA ATG CAA	607
*Lr39*	1D	GDM35	CCT GCT CTG CCC TAG ATA CG ATG TGA ATG TGA TGC ATG CA	190
*Lr47*	7A	PS10	GCT GAT GAC CCT GAC CGG T TCT TCA TGC CCG GTC GGG T	282

## Data Availability

Data are contained within the article.

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
