# Peer review of "Genetic Control of Effective Seedling Leaf Rust Resistance in Aegilops biuncialis Vis. Accessions from the VIR Collection"

_plants, 2024, doi:10.3390/plants13162199_

Round 1

Reviewer 1 Report

Comments and Suggestions for Authors

Comments on the Quality of English Language

Author Response

Comments 1. However, the approach using a mixed Pt population to phenotype the segregating populations is a major flaw of the genetic analysis, which could potentially impact the validity of the results. Based on the infection types described by the authors, the resistances conferred by the selected Ae. biuncialis accessions are most likely race-specific, meaning they follow the gene-for-gene hypothesis; each Lr gene has its corresponding Avr gene. Using a mixed Pt population equates to using a pool of variable and mixed Avr genes for the genetic analysis; it not only complicates the segregation ratio but is also incorrect. The authors should use Pt isolates for the infection-type phenotyping

Response 1. The main and in fact single task of the study was to reveal genetic control of effective leaf rust resistance (expressing against all or almost all genotypes of the pathogen) in Ae. biuncialis accessions. It was the reason to use mixture of the pathogen genotypes to infect the plants. In the case of Pt isolates use we would likely reveal many genes for resistance. We absolutely agree with the reviewer opinion that in population  a pool of variable and mixed Avr genes exists, and for different isolates we would have different segregations and consequently different numbers of genes for the resistance. And this information is of no scientific interest. Moreover, we could find genes for resistance in accessions susceptible to Pt population, but this information is of no use.

To note, study of segregation for resistance under field conditions (in the case of severe disease development) is often used by breeders to study genetics of the trait and new varieties development. Evidently in this case there is mixture of pathogen genotypes.

Comments 2.  In addition, testing if the selected Ae. biuncialis have the same genes as the five known Lr genes is unnecessary because these Lr genes were identified from different species. Instead, allelic analysis to test if the Lr genes from the selected Ae. biuncialis accessions are on the same locus or different loci would be useful information for future alien gene integration and application in breeding.

Response 2. The reviewer is absolutely right. Five genes were transferred from different species. But there are many publications concerning identification of genes for resistance in species different from original sources of these genes. And 4 references are presented in this study. As for usefulness of information about the same or different loci, the reviewer again is absolutely right. And we did perform allelic analysis (2.2.1, second part).

Comments 3. Authors should use correct scientific terminologies in their descriptions. For examples. “Seedling Stage” for plants, not “Juvenile Stage”; “Accessions” not “samples”; “Infection type” not “type of reaction” etc.

Response 3. We changed. But infection type as synonym for type of reaction is widely used in English literature.

Comments 4. The calculations in Table 2 do not add up.

Response 4. Thanks for this remark. Really it was misprint in number of resistant plants in one cross. We checked.

Comments 5. Many descriptions in the text are confusing. For example, in lines 272-273, “The sample k-3003 (Russia) (Figure 1) served as heavily affected by Pt parent and control.”

Response 5. We tried to check and eliminate these types of mistakes.

We are very grateful to the reviewer for attentive reading of the article and very valuable remarks.

Reviewer 2 Report

Comments and Suggestions for Authors

General Comments

The manuscript describes a study on leaf rust resistance in eight accessions of Aegilops biuncialis. The study suggests that these eight resistant accessions likely possess the same Lr resistance gene, which differs from a list of five Lr genes effective in certain regions of the Russian Federation. This research identifies a potentially new source of resistance to wheat leaf rust.

The study employs a basic crossing and Mendelian approach to estimate the number of genes involved in the resistance among the eight Aegilops biuncialis accessions and to explore their relationship to other Lr genes. However, it does not attempt to map the gene to a chromosomal location. While the experiments described are scientifically sound, the manuscript may not meet the standards expected in the third decade of the 21st century without additional advancements. Additionally, the text requires extensive English editing for clarity and readability.

Specific Comments

Passport Data of the Accessions: Please provide detailed passport data for the Aegilops biuncialis accessions used in the study.

Line 130: The text in this section is difficult to read. It would be clearer to present this information in a table format, along with a concise description of the general conclusions drawn from the crossing experiments.

Line 151 (PCR Analysis): The markers used in your PCR analysis are derived from other alien species. These markers are effective only within the bread wheat gene pool where these genes have been introgressed, or in the original species. They are not suitable for locating the gene in different backgrounds. Consider this limitation in your analysis and discussion.

Tables 2 and 3: It appears that all the crossings exhibit similar segregation patterns. Perform a chi-square test to evaluate the hypothesis that all ratios originate from the same distribution. Note that the sample counts are too small to reliably differentiate between various hypothetical segregation patterns.

Lines 322-324: This section appears to be incorrect and needs a better explanation. Additionally, the entire paragraph requires more details about the resistance test protocol.

Summary

While the study provides useful insights into leaf rust resistance in Aegilops biuncialis, the manuscript would benefit from additional data and a clearer presentation. Addressing the limitations of the PCR markers used, conducting appropriate statistical tests on the crossing data, and providing a more detailed explanation of the resistance test protocol will strengthen the conclusions. Finally, extensive English editing is necessary to improve the readability and clarity of the manuscript.

Comments on the Quality of English Language

Extensive English editing is necessary to improve the readability and clarity of the manuscript.

Author Response

Comments 1. The study employs a basic crossing and Mendelian approach to estimate the number of genes involved in the resistance among the eight Aegilops biuncialis accessions and to explore their relationship to other Lr genes. However, it does not attempt to map the gene to a chromosomal location. While the experiments described are scientifically sound, the manuscript may not meet the standards expected in the third decade of the 21st century without additional advancements. Additionally, the text requires extensive English editing for clarity and readability.

Response 1. The general purpose of the study was to study genetics of effective Pt resistance in accessions of goat grass in view of possible future introgression into wheat genome. From our viewpoint chromosomal localization of gene for Pt resistance at this stage of the work is of no interest in genetics and breeding. It could be interested to localize the gene after its introgression  into wheat genome. Last years, many works are published in identification and genetic study of resistance in wild relatives of wheat without chromosomal localization of corresponding genes. Some of these works are referred in the article. As for English we sent the article to 3 English speaking persons and took into account all their comments.

Comments 2. Passport Data of the Accessions: Please provide detailed passport data for the Aegilops biuncialis accessions used in the study.

Response 2. We added this information as a separate table.

Comments 3. Line 130: The text in this section is difficult to read. It would be clearer to present this information in a table format, along with a concise description of the general conclusions drawn from the crossing experiments.

Response 3. It would be very strange table. Very narrow (only 2 columns) and very long. And all conclusions are in Discussion section.

Comments 4. Line 151 (PCR Analysis): The markers used in your PCR analysis are derived from other alien species. These markers are effective only within the bread wheat gene pool where these genes have been introgressed, or in the original species. They are not suitable for locating the gene in different backgrounds. Consider this limitation in your analysis and discussion.

Response 4. The reviewer is absolutely right. But we have some thoughts. The molecular markers are not effective in many cases even in bread or in the original species (references 56, 69, 70). But there are many publications about identification of the genes in different backgrounds (41, 67-69, Hovhannisyan, N.A.  et al. 2011,  Gultiaeva et al., 2014,  Davoyan et al., 2012). So, we used this method to get some additional information about the identified gene. And we give information on the impossibility to identify Lr9 in Ae. biuncialis (63).

Comments 5. Tables 2 and 3: It appears that all the crossings exhibit similar segregation patterns. Perform a chi-square test to evaluate the hypothesis that all ratios originate from the same distribution. Note that the sample counts are too small to reliably differentiate between various hypothetical segregation patterns.

Response 5.  1. Really, we have similar segregation in all crosses and now we know why. All accessions have one identical dominant gene for resistance. To get this information we did the work. 2. How it is possible to combine data for different cross combinations (they are result of crossing of different genotypes)? 3. We did write that number of plants in F2 is too small to differentiate between various hypothetical segregation patterns (to distinguish 3:1 and 49:15 we must have and evaluate more than 5000 F2 plants, for Aegilops it is quite difficult) that why we analyzed F3 too.

Comments 6. Lines 322-324: This section appears to be incorrect and needs a better explanation. Additionally, the entire paragraph requires more details about the resistance test protocol.

Response 6. We corrected  the paragraph.

 Comments 7. Summary

While the study provides useful insights into leaf rust resistance in Aegilops biuncialis, the manuscript would benefit from additional data and a clearer presentation. Addressing the limitations of the PCR markers used, conducting appropriate statistical tests on the crossing data, and providing a more detailed explanation of the resistance test protocol will strengthen the conclusions. Finally, extensive English editing is necessary to improve the readability and clarity of the manuscript.

Response 7.  As for clearer presentation we tried to change the text according to  comments of 3 English speaking persons. In Discussion we presented data of limitations of the PCR markers used with additional references. From our viewpoint chi-square test is appropriate statistical test on the crossing data and one misprint mistake was corrected. We tried give detailed explanation of the resistance test protocol.

Comments 8. Extensive English editing is necessary to improve the readability and clarity of the manuscript.

Response 8. We sent the article to we sent the article to 3 English speaking persons and took into account all their comments.

We are very grateful to the reviewer for attentive reading of the article and very valuable remarks.

Reviewer 3 Report

Comments and Suggestions for Authors

Kolesova and Tyryshkin studied genetic control of the resistance to leaf rust in Ae. biuncialis species employing hybridological, phytopathological, and molecular analyses. The manuscript provides some information. However, there are few crucial points that if considered will increase the value of the manuscript and may be readability.

-In the abstract, please modify the sentence ‘The cultivated…the trait’. ‘and wild wheats’ can be removed.

-English language is a major issue at different point. Please check the entire manuscript for such mistakes. For example, Line-24-25. Line 53-55. Line 269-270. Line 292, Line 299. Please rewrite such sentences.

- In the introduction, please elaborate the importance of Aegilops biuncialis citing some research based studies where the species has been studied for biotic and abiotic stress tolerance. For example; https://doi.org/10.3390/biology11081094; https://doi.org/10.3389/fpls.2021.736614; https://doi.org/10.1071/FP03143; https://doi.org/10.1038/s41598-020-79372-1  

-Please explain about the cultivar ‘Leningradka’ in the material section. From where, it was collected?

-Line 294- Seedlings were sown in the field or in greenhouse pots?

-Line 312- What about the analysis after the second time infection?

-In 4.4. Phytopathological Analysis- Which samples were infected, when and where? It will be better to write plants instead of samples.

-Line 333-Please add the details of the PCR conditions to the manuscript.

-Tables and Figure captions must be self explanatory. For example, Table 2 caption, in which crop?

-Which 100 bp DNA ladder was used?

I do believe that the manuscript can be accepted once the authors address the mentioned points and enrich the manuscript with the crucial information. 

Comments on the Quality of English Language

Extensive editing of English language required

Author Response

Comments 1. In the abstract, please modify the sentence ‘The cultivated…the trait’. ‘and wild wheats’ can be removed.

Response 1. We did it. But in fact, most accessions of wild wheats are susceptible to the rust.

Comments 2. English language is a major issue at different point. Please check the entire manuscript for such mistakes. For example, Line-24-25. Line 53-55. Line 269-270. Line 292, Line 299. Please rewrite such sentences.

Response 2. We tried to do it.

Comments 3. In the introduction, please elaborate the importance of Aegilops biuncialis citing some research based studies where the species has been studied for biotic and abiotic stress tolerance. For example; https://doi.org/10.3390/biology11081094; https://doi.org/10.3389/fpls.2021.736614; https://doi.org/10.1071/FP03143; https://doi.org/10.1038/s41598-020-79372-1

Response 3. We added these references to discussion.

Comments 4. Please explain about the cultivar ‘Leningradka’ in the material section. From where, it was collected?

Response 4. It was done.

Comments 5. Line 294- Seedlings were sown in the field or in greenhouse pots?

Response 5. Some F1 seedlings of each cross combinations were sown in the field and others in greenhouse pots to ensure in getting seeds.

Comments 6. Line 312- What about the analysis after the second time infection?

Response 6. We added this information to the text.

Comments 7. In 4.4. Phytopathological Analysis- Which samples were infected, when and where? It will be better to write plants instead of samples.

Response 7. We added information. We deleted the word ‘sample’ over the text.

Comments 8. Line 333-Please add the details of the PCR conditions to the manuscript.

Response 8. We added this information.

Comments 9. Tables and Figure captions must be self explanatory. For example, Table 2 caption, in which crop?

Response 9. We checked it. In Table 2 caption it is written Ae. biuncialis.

Comments 10. Which 100 bp DNA ladder was used?

Response 10. We added this information.

We are very grateful to the reviewer for attentive reading of the article and very valuable remarks.

Reviewer 4 Report

Comments and Suggestions for Authors

Review of the manuscript titled “Genetic Control of Effective Seedling Leaf Rust Resistance in Aegilops biuncialis Vis. Samples from the VIR Collection”.

The study is devoted to an important topic of identification of new leaf rusts resistance genes in wild species. The authors use unique material, relevant methodology and arrive at logical results and conclusions. The study contributes to our knowledge and deserves publication. However, there are some options for improvement.

1.        While describing the study material resistance to leaf rust, the authors make reference to two their own publications 10 and 47. Publication 10 is not available at Google Scholar and publication 47 is in Russian. This is why it would be good to describe more in detail the adult plant resistance of these six accessions in the paper instead of giving the references. Since the adult plant resistance was evaluated on leaves in chambers and published in 2017 – may be authors have more recent field data showing reaction of this material to leaf rust.

2.        The authors present the use of oligo genes and seedling type of resistance as the most effective strategy to protect against leaf rust. Many researchers do not share such an approach. That is why the authors can also mention other strategies such as adult plant resistance, durable resistance concept, combination of minor genes, pyramiding, etc. And how the newly discovered gene fits in different breeding strategies.

3.        Discussion first paragraph repeats the introduction. Overall, discussion sounds more like repetition of the results. No need to have references to tables and figures in discussion section. One important topic is not covered in discussion – crosses of Aegilops biuncialis with wheat. How easy are these crosses and if any germplasm was developed from these crosses and for which traits. What would be the crossing and leaf rusts resistance gene introduction strategy.

4.        English requires  careful reading to correct several mistakes like line 49 “That why” and others. “Samples” in reference to genotypes may be best described by “accessions’ or “entries”. “Hybrids” meaning in this paper is “populations”.

Comments on the Quality of English Language

Included in review..

Author Response

Comments 1.  While describing the study material resistance to leaf rust, the authors make reference to two their own publications 10 and 47. Publication 10 is not available at Google Scholar and publication 47 is in Russian. This is why it would be good to describe more in detail the adult plant resistance of these six accessions in the paper instead of giving the references. Since the adult plant resistance was evaluated on leaves in chambers and published in 2017 – may be authors have more recent field data showing reaction of this material to leaf rust.

Response 1. The task of the present work was to give genetic characterization of effective seedling resistance to leaf rust in 8 accessions of goatgrass. So, we think the information on adult resistance is not absolutely necessary (genotypes with seedling resistance often are resistant in flag leaf stage). But we add the short information  in Introduction section.

Unfortunately, we have to refer to published articles with data used in present work. Therefore, we had to use Russian references.

Comments 2. The authors present the use of oligo genes and seedling type of resistance as the most effective strategy to protect against leaf rust. Many researchers do not share such an approach. That is why the authors can also mention other strategies such as adult plant resistance, durable resistance concept, combination of minor genes, pyramiding, etc. And how the newly discovered gene fits in different breeding strategies.

Response 2. Unfortunately we could not find where we presented comparisons of effectiveness of different strategies, cause it was not a task of the work.

Comments 3. Discussion first paragraph repeats the introduction. Overall, discussion sounds more like repetition of the results. No need to have references to tables and figures in discussion section. One important topic is not covered in discussion – crosses of Aegilops biuncialis with wheat. How easy are these crosses and if any germplasm was developed from these crosses and for which traits. What would be the crossing and leaf rusts resistance gene introduction strategy.

Response 3. First paragraph Discussion (6 lines) briefly explained why for it was necessary to do the work. From our viewpoint we gave results without discussion in Results and then discussed them in Discussion in limits of the assigned purpose. Crossability of Aegilops biuncialis with wheat is absolutely out of the article field. Evidently to cross species from tertiary genepool is not as easy as species from primary one. But it is not impossible (41, 40, 54, 59). We deleted references for tables and figures

Comments 4.   English requires  careful reading to correct several mistakes like line 49 “That why” and others. “Samples” in reference to genotypes may be best described by “accessions” or “entries”. “Hybrids” meaning in this paper is “populations”.

Response 4.

We changed “That why”. We deleted the word ‘sample’ over the text. The word “Hybrids” should be understood in the classical sense.

 We are very grateful to the reviewer for attentive reading of the article and very valuable remarks.

Round 2

Reviewer 1 Report

Comments and Suggestions for Authors

I agree with the authors that infecting plants with a mixture of the pathogen genotypes can reveal all effective resistance genes in Ae. biuncialis. However, when it comes to genetic analysis, it is important to know how many different genes confer resistance in each accession and their specificity. This information is the basis for using those resistant accessions. The authors stated that “ … and for different isolates we would have different segregations and consequently different numbers of genes for the resistance. And this information is of no scientific interest. Moreover, we could find genes for resistance in accessions susceptible to Pt population, but this information is of no use”.

Besides, the description of the results is vague. To make it simple, after the analysis, can the authors conclude how many different genes were identified from the selected Ae. biuncialis accessions? What are the natures of the Lr genes?

Comments on the Quality of English Language

Need improvement

Author Response

Comments 1: I agree with the authors that infecting plants with a mixture of the pathogen genotypes can reveal all effective resistance genes in Ae. biuncialis. However, when it comes to genetic analysis, it is important to know how many different genes confer resistance in each accession and their specificity. This information is the basis for using those resistant accessions. The authors stated that “ … and for different isolates we would have different segregations and consequently different numbers of genes for the resistance. And this information is of no scientific interest. Moreover, we could find genes for resistance in accessions susceptible to Pt population, but this information is of no use”.

Response 1: Thanks to the Reviewer 1. We studied EFFECTIVE resistance expressing against all the pathogen isolates in populations. If we had found 5 genes for resistance to isolate X but four of them would have been ineffective against other isolates, is this information interested. The article is under preparation where I will show that any accession is resistant to certain isolates under specific conditions.

Comments 2: Besides, the description of the results is vague. To make it simple, after the analysis, can the authors conclude how many different genes were identified from the selected Ae. biuncialis accessions? What are the natures of the Lr genes?

Response 2: Again thanks. We wrote several times across the article that we found only one gene for effective resistance. The biochemical and molecular nature of the gene was absolutely out of the limits of the article. Possibly it the task of future study.

Reviewer 2 Report

Comments and Suggestions for Authors
  1. Response 3:

    I do not accept this response. It is straightforward to create an 8x8 table with the parents listed in both columns and rows and to write the R:S ratio in the appropriate cell.

    2. Response 5:

    If you agree that the numbers are low, I see no reason to test other ratios than 1:3 and load the tables with redundant data. You can conduct a Fisher exact test between pairs of ratios for each cross to see if all come from the same distribution, meaning they have the same combination of genes.

    3. I am not convinced that reporting a single resistance gene without any attempt to locate it on the wheat genome is of interest to the wheat community. This gene could be allelic to other U or M genome Lr genes.

    4. Line 60-61: The taxonomic definition should be clearer and less complex.

    5. Line 73: This section needs a smoother transition, as it starts abruptly.

    6. Response 6: You still need to indicate the number of replicates for each experiment.

Comments on the Quality of English Language
  1. The level of English has improved but still requires further editing by a native English speaker. For example, use "wheat" instead of "wheats".

Author Response

Comments 1:

Response 3:

I do not accept this response. It is straightforward to create an 8x8 table with the parents listed in both columns and rows and to write the R:S ratio in the appropriate cell.

Response 1: We added this table.

Comments 2:

Response 5:

If you agree that the numbers are low, I see no reason to test other ratios than 1:3 and load the tables with redundant data. You can conduct a Fisher exact test between pairs of ratios for each cross to see if all come from the same distribution, meaning they have the same combination of genes.

Response 2: Thanks a lot to the Reviewer 2. As we know chi-square test can be used when the number of plants in all classes is bigger than 10 (our cases). Why it is impossible to check 13:3 if we have 100 plants? If we have two crosses ThatcherLr9 x Thatcher and ThatcherLr19 x Thatcher and Fisher exact test doesn’t find statistical differences in segregation, does it mean we have the same combinations of genes? In total the table 2 is the evidence to obtain and study F3 generation.

Comments 3: I am not convinced that reporting a single resistance gene without any attempt to locate it on the wheat genome is of interest to the wheat community. This gene could be allelic to other U or M genome Lr genes.

Response 3: U genome effective Lr genes  are Lr9 and 76. Identified gene is not Lr9 according to test clone analysis and partially to PCR product visualization. It is not Lr 76 because of absolutely different phenotypic expression (M. Bansal et al., 2020; M. Bansal et al., 2017). M genome Lr genes are Lr57 and 62. Lr 57 is not effective and Lr 62 (Marais, et al., 2009; Antonov and Marais, 1996) has a very different phenotypic expression.

Chromosome location in wheat genome does not guarantee the nonallelic state (as it was for Lr19). If to read the list of references, it could be seen many works on resistance in wheat relatives without chromosome localization, but they are interesting for wheat community.

Comments 4: Line 60-61: The taxonomic definition should be clearer and less complex.

Response 4: It is impossible to change taxonomic definition.

Comments 5: Line 73: This section needs a smoother transition, as it starts abruptly.

Response 5: We did it.

Comments 6: Response 6: You still need to indicate the number of replicates for each experiment.

Response 6: We wrote that each hybrid population were inoculated twice. Parental lines were tested in dozens of experiments as they were used as controls.

Comments 7: The level of English has improved but still requires further editing by a native English speaker. For example, use "wheat" instead of "wheats".

Response 7: We sent the text again to native English speaker.

Reviewer 3 Report

Comments and Suggestions for Authors

Authors have modified the manuscript as per the provided suggestions. 

Comments on the Quality of English Language

Minor editing of English language required.

Author Response

Comments 1: Authors have modified the manuscript as per the provided suggestions. 

Response 1: Thanks a lot to the Reviewer 3.

Comments 2: Minor editing of English language required.

Response 2: We did it.

Round 3

Reviewer 1 Report

Comments and Suggestions for Authors

no more comments

Comments on the Quality of English Language

No more comments